# Expression Profile of miR-199a and Its Role in the Regulation of Intestinal Inflammation

**DOI:** 10.3390/ani13121979

**Published:** 2023-06-14

**Authors:** Zijuan Wu, Yanyun Yan, Wenli Li, Yali Li, Huansheng Yang

**Affiliations:** Hunan Provincial Key Laboratory of Animal Intestinal Function and Regulation, Hunan Normal University, No. 36 Lushan Road, Changsha 410081, China; wuzijuan2000@163.com (Z.W.); liyouyou06@gmail.com (Y.Y.); liyali06@163.com (W.L.)

**Keywords:** miR-199a, weaning stress, piglets, colitis, inflammation

## Abstract

**Simple Summary:**

Early weaning stress impairs intestinal epithelial barrier, causing immune system dysfunction and growth retardation. In the context of inflammation, microRNAs (miRNAs) are crucial for maintaining host homeostasis, while their implication for animal health and performance remains unclear. Therefore, the profiles and roles of miRNAs in piglets during weaning stress were explored in this study. Results demonstrated an altered expression profile of miR-199a in postweaning piglets and a regulatory role for miR-199a in intestinal inflammation, suggesting that miR-199a may possess the potential to be used as novel biomarkers and anti-inflammatory agents. These findings may provide insight into new therapeutic intervention for animal health and production.

**Abstract:**

Early weaning stress impairs intestinal health in piglets. miRNAs are crucial for maintaining host homeostasis, while their implication for animal health remains unclear. To identify weaning-associated miRNAs, piglets were sampled at day 0, 1, 3, 7 and 14 after weaning. The data indicated that the highest levels of miR-199a-5p in jejunal villus upper cells were observed on day 14 after weaning, while the lowest levels in crypt cells were noted on day 7 and 14. In contrast, miR-199a-3p was down-regulated in both of these two cells on day 7 after weaning compared with day 0. Both miR-199a-5p and -3p were differently expressed along the villus–crypt axis. To further clarify the function of miR-199a, mice deficient in miR-199a were exposed to dextran sulfate sodium (DSS) to induce colitis. Results revealed that silencing of miR-199a enhanced sensitivity to DSS-induced colitis. Moreover, the increased morbidity and mortality were correlated with enhanced inflammatory cell infiltration, elevated pro-inflammatory cytokine expression, impaired barrier function, and a concomitant increase in permeability-related parameters. Bioinformatic analysis further demonstrated that lipid metabolism-related pathways were significantly enriched and Ndrg1 was verified as a target of miR-199a-3p. These findings indicate that miR-199a may be important for animal health management.

## 1. Introduction

MicroRNAs (miRNAs) are small non-coding RNAs involved in post-transcriptional gene regulation via facilitating messenger RNA (mRNA) degradation or translational repression [1,2]. They have been implicated in various developmental processes, such as embryonic, neuronal, and immune system development [3]. Accumulating evidence has shown that miRNA mutations or aberrant expression patterns correlate with uncontrolled inflammatory processes, indicating a crucial role in the maintenance of immune homeostasis [4]. Differential miRNA expressions have been noted in the inflamed colonic mucosa of patients with inflammatory bowel disease, and identified as biomarkers for diagnosis, prognosis, and personalized therapy [5,6,7]. miRNAs hold great potential as regulators, molecular markers and therapeutic targets for human diseases, while their applications for livestock production remain elusive [8,9,10].

Weaning is a pivotal transition phase in swine production. Stress associated with early weaning in piglets is often characterized by enhanced mucosal immune activation and impaired barrier function [11,12]. Weaning stress also perturbs cellular differentiation along the crypt–villus axis, accompanied by villus shortening and crypt deepening [13]. Therefore, exploration of miRNAs as monitoring markers for stress-associated disorders may provide a new health management approach for livestock [14,15,16]. A previous study revealed that a series of miRNAs in the small intestine between suckling and weaning piglets expressed differently after weaning [17]. Recently, another study identified 38 weaning-associated miRNAs in piglets before and after weaning [18]. However, the potential function of miRNAs in intestinal injury and inflammation control during weaning process still requires further elucidation. Aberrant expression of miR-199a-5p and -3p, both of which are produced from pre-miR-199a, has been found in the peripheral blood of patients with active colitis [10,19] and in animal colitis models [20]. Moreover, miR-199a-5p and -3p have been reported to be upregulated in jejunal villus cells in suckling piglets compared with crypt cells [21]. Herein, the dynamic expression profiles of miR-199a-5p and -3p were detected at day 0, 1, 3, 7, and 14 after weaning to evaluate their potentiality as biomarkers associated with weaning stress. In general, miRNAs show high sequence conservation across species, and the biological functions of miRNAs can be better understood by specific gene-deletion approaches [3]. In this study, mice deficient in miR-199a were exposed to an experimental DSS-induced colitis to further explore the role of miR-199a in the maintenance of immune balance.

## 2. Materials and Methods

### 2.1. Isolation of Jejunal Epithelial Cells

A total of 20 piglets (Duroc × (Landrace × Large Yorkshire)) with similar bodyweight were weaned at 21 days old. Piglets were sacrificed at day 0, 1, 3, 7, and 14 after weaning (4 animals at each time-point). Protocols for isolating piglet intestinal epithelial cells were described previously [22,23]. Briefly, segments of mid-jejunum were dissected and washed with phosphate buffered saline (PBS), then incubated at 37 °C for 30 min with oxygenated PBS buffer. Following a 40 min incubation in oxygenated EDTA chelating buffer, the buffers were centrifuged three times at 1000× rpm for 10 min for sequential collection of jejunal villus cells and crypt cells. Cells were then rinsed with suspension buffer (10 mM HEPES, 1.5 mM CaCl_2_ and 2.0 mM MgCl_2_), centrifuged, and stored at −80 °C.

### 2.2. Genotype Identification

Wild-type C57BL/6 mice were purchased from Hunan Silaike Jingda Experimental Animal Co., Ltd. (Changsha, China). Female miR-199a-deficient mice (C57BL/6 background, CRISPR/cas9-system) were obtained from Nanjing Biomedical Research Institute (Nanjing, China). Mice were housed in a standard specific-pathogen-free (SPF) facility (Changsha, China). Mouse tail genomic DNA was extracted via Genomic DNA Extraction Kit (Takara, Dalian, China) and genotyping was analyzed by PCR. Primers used to identify C57BL/6 wild-type (WT) and miR-199a knockout (KO) mice were as follows: miR-199a F: 5′-TCCAGATGTGAGCAAGTGGC-3′; R: 5′-CCAAGATAAAGACCAGCAGG-3′. PCR amplification parameters were set as 30 cycles of 98 °C for 10 s, 55 °C for 30 s, and 72 °C for 60 s. The PCR-amplified products were further confirmed by DNA sequencing.

### 2.3. DSS Treatment

To induce colitis, six- to eight-week-old female WT or miR-199a KO mice were treated with normal water or 4% DSS (MP Biomedical, Santa Ana, CA, USA) for 7 days. Mice were monitored daily for morbidity and mortality. Disease activity index (DAI) was assessed by scoring the changes in body weight, diarrhea, and bleeding scores as defined previously [24]. After 7 days of the DSS challenge, serum, colon, and spleen samples were taken for subsequent experimental analysis.

### 2.4. RNA Extraction and Real-Time PCR

Total RNA from mouse colonic tissues was isolated with RNAiso Plus (Takara, Dalian, China). RNA quality was evaluated using agarose gel electrophoresis. Subsequently, RNA samples were converted to cDNA using a reverse transcription kit (Takara, Dalian, China). Next, cDNA samples were examined for tumor necrosis factor α (TNF-α), interleukin-1β (IL-1β), interleukin-6 (IL-6), interleukin-23 (IL-23), Claudin-1, Occludin, and N-myc downstream regulated gene 1 (Ndrg1) expression. Data were normalized using β-actin as a reference gene. Primers were designed using the NCBI Primer-BLAST tool (https://www.ncbi.nlm.nih.gov, last accessed on 1 May 2023). Primers used in the present study are listed in Table 1. To test the expression levels of miR-199a-5p and -3p, small RNA was reverse transcribed via miRNA 1st-Strand cDNA Synthesis Kit (Accurate Biotechnology, Changsha, China). Forward primer sequences for miR-199a-5p were as follows: 5′-CCCAGTGTTCAGACTACCTGTTC-3′; for miR-199a-3p: 5′-ACAGTAGTCTGCACATTGGTTA-3′. Reverse sequences and U6 (small nuclear RNA) primers were included in the kit. Real-time PCR reactions were performed on a Quanstudio 5 instrument (Thermo Fisher, Singapore). All samples were analyzed simultaneously in triplicate. Relative expression was analyzed using the 2^−ΔΔCT^ method.

### 2.5. Histopathological Evaluation

Colonic and splenic tissues were fixed in a 10% formalin, embedded in paraffin, and then sectioned. Slices were stained with hematoxylin and eosin according to standard protocol. Images were taken using a Leica DM3000 microscope (Wetzlar, Germany). Inflammation and tissue damage were assessed in a blinded manner. Colonic inflammation was scored as a combination of inflammatory cell infiltration (score range: 0–4) and intestinal architecture disruption (score range: 0–4) [24]. Splenic inflammation was graded as 0 (normal), 1 (minimal), 2 (mild), 3 (moderate), and 4 (marked) as described in Appendix A.

### 2.6. Intestinal Permeability Test

Mice were fasted for 4 h; then, they received a gavage of fluorescein isothiocyanate (FITC)-dextran from Sigma (St. Louis, MO, USA) at a dose of 600 mg/kg bodyweight. At 4 h after gavage, blood samples were centrifuged at 3000 rpm for 10 min to obtain the plasma. FITC-dextran signal was examined at 490/525 nm using a fluorescence microplate detector (Biotek, Rochester, VT, USA).

### 2.7. Immunofluorescence Staining

Paraffin-embedded colon tissues were sectioned at 4 μm of thickness using a Leica microtome (RM2235, Nussloch, Germany). After dewaxing, hydration and antigen retrieval, the sections were incubated with a blocking buffer from Vector Laboratories (Burlingame, CA, USA). Then, the macrophages were immune-stained with FITC-labeled F4/80 antibodies (eBioscience, San Diego, CA, USA) and the neutrophils were stained with PE/Cyanine5-conjugated Gr-1 antibodies (Biolegend, San Diego, CA, USA). Cell nuclei were stained using an anti-fluorescence quenching sealing solution with DAPI (Vector Laboratories) and the sections were examined by a Leica fluorescence microscope (Wetzlar, Germany). Gr1^+^ cells and F4/80^+^ cells were quantified by counting four high-power fields on each section.

### 2.8. Serum Biochemical Analysis

Blood samples were collected prior to execution, and serum was obtained by centrifugation at 3000 rpm for 10 min. Triglyceride, total cholesterol, high-density lipoprotein, and low-density lipoprotein in serum were measured using a Cobas C311 biochemical analyzer (Roche Diagnostics, Rotkreuz, Switzerland).

### 2.9. ELISA Assay

Concentrations of lipopolysaccharide (LPS), diamine oxidase (DAO), and D-lactate (D-LA) in serum were detected using ELISA kits from eBioscience (San Diego, CA, USA) according to the manufacturer’s protocol.

### 2.10. Bioinformatics Analysis

For RNA-sequencing, extraction of total RNA from mouse colonic tissues was performed using TRIzol reagents (Invitrogen, Carlsbad, CA, USA). RNA samples with an RIN value > 9.0 were used to construct sequencing libraries with Illumina TruseqTM RNA Kit. Paired-end (PE, 2 × 150 bp) sequencing was performed on an Illumina NovaSeq 6000 from Majorbio (Shanghai, China). After check for read quality with FASTP, clean reads remained and were mapped to mouse genome [24]. Differential expression analysis was carried out using DESeq2. Genes with |log2FC| ≥ 1 and adjusted *p* < 0.05 were defined as significantly differentially expressed genes (DEGs). Gene Ontology (GO) analysis was performed using Goatools with Bonferroni-corrected *p* < 0.05. Kyoto Encyclopedia of Genes and Genomes (KEGG) and Reactome were carried out by Metascape (v3.5.20230501) (https://metascape.org/, accessed on 3 May 2023) as previously described [24]. Prediction of target genes of miR-199a-5p or -3p was performed by StarBase v2.0 (https://starbase.sysu.edu.cn, accessed on 1 May 2023).

### 2.11. Luciferase Reporter Assay

DNA fragments (801 bp) containing the putative miR-199a-3p binding site in the 3′ UTR (3′ untranslated region) of Ndrg1 (wild-type, WT) and its mutants were cloned into the pmiR-RB-REPORT™ vector from RiboBio (Guangzhou, China). Vectors, negative controls or mimics-miR-199a-3p (designed and synthesized by RiboBio, Guangzhou, China) were co-transfected into HEK293T cells using the Lipofectamine 2000 reagent from Invitrogen (Carlsbad, CA, USA). Cells were collected and luciferase activity was determined 48 h after transfection using the Dual Luciferase Reporter Assay Kit (Promega, WI, USA). The Renilla luciferase signal was normalized over the Firefly luciferase signal.

### 2.12. Statistical Analysis

Results are presented as the mean ± SEM. Differences between the survival curves were compared by log-rank test. *p* values were calculated using unpaired Student’s *t* test, Mann–Whitney test, or one-way ANOVA with Tukey’s post hoc test with a GraphPad Prism 7.0 (San Diego, CA, USA). A *p*-value < 0.05 was considered statistically significant.

## 3. Results

### 3.1. Expression Levels of miR-199a in Piglets during the Postweaning Period

As shown in Figure 1A,B, the highest expression levels of miR-199a-5p in jejunal villus upper cells were observed on day 14 after weaning while the lowest levels in crypt cells were noted on day 7 and 14 after weaning (*p* < 0.05). Moreover, it was found that miR-199a-5p was significantly up-regulated in jejunal villus cells compared with crypt epithelial cells on day 14 after weaning (*p* < 0.05), with the highest expression ratio (Figure 1C).

In addition, miR-199a-3p expression levels in jejunal villus cells decreased significantly on day 3 and 7 after weaning when compared to day 0 while it increased significantly on day 14 after weaning (Figure 1D). Notably, the levels of miR-199a-3p were also significantly down-regulated on day 7 after weaning in crypt cells (*p* < 0.05) (Figure 1E). As demonstrated in Figure 1F, the greatest ratio of miR-199a-3p, similarly to that of miR-199a-5p, was observed on day 14 after weaning compared with all other time-points.

### 3.2. Loss of miR-199a Increased Susceptibility to DSS-Induced Colitis

After confirming the homozygous KO mice by PCR (Figure 2A), a DSS-induced colitis model was established by administering 4% DSS in drinking water. The expression levels of miR-199a-5p and -3p were up-regulated in colonic tissues of WT mice after DSS treatment (Appendix A). In contrast to WT mice, miR-199a-deficient mice exhibited more severe colitis, with worse survival (*p* < 0.05; Figure 2B), dramatic weight loss (on day 3 and 6, *p* < 0.05; on day 7, *p* < 0.01; Figure 2C), greater disease activity index (Figure 2F), as well as significant colon shortening (Figure 2G,H) after DSS treatment. However, no significant differences were observed for diarrhea score (Figure 2D) and bleeding score (Figure 2E). Histological analysis revealed worse colon inflammation in miR-199a KO mice, as evidenced by severe crypt loss, extensive mucosal damage, marked submucosal edema, and massive infiltration of inflammatory cells throughout the entire intestinal wall (Figure 2I; Appendix A). Moreover, extensive pathological changes in the spleen tissues, with significantly decreased white pulp and increased red pulp, were observed in miR-199a KO mice following DSS challenge (Figure 2J; Appendix A).

### 3.3. Deficiency of miR-199a Exacerbated Immune Response in Mice with DSS Challenge

Significantly increased F4/80^+^ macrophage and Gr-1^+^ neutrophil infiltration was found in the colon of miR-199a-deficient mice after DSS administration (Figure 3A,B; Appendix A). Consistently, enhanced levels of TNF-α, IL-1β, and IL-23 were observed in miR-199a KO mice treated with DSS compared to mice given normal drinking water (Figure 3C,D,F). Furthermore, ablation of miR-199a significantly up-regulated the expression level of TNF-α when compared with WT mice after the DSS challenge (*p* < 0.01).

### 3.4. Impaired Intestinal Barrier Function in miR-199a Knockout Mice following DSS Treatment

As presented in Figure 4A, deficiency of miR-199a significantly elevated intestinal permeability after DSS challenge (*p* < 0.01), indicating reduced colonic barrier function in miR-199a-deficient mice. Consistently, higher concentrations of LPS, DAO, and D-LA were found in mice treated with DSS (Figure 4B–D). Notably, higher concentrations of LPS and D-LA were noted in mice lacking miR-199a when compared with WT mice in response to the DSS challenge. After DSS treatment, increased expression levels of Occludin were observed in both strains (Figure 4F) while up-regulated Claudin-1 was only found in WT mice (Figure 4E).

### 3.5. Identification of Key Pathways in Mice Lacking miR-199a

A total of 264 genes were found to show altered expression profile between miR-199a-deficient mice and WT mice with colitis, including 154 up-regulated and 110 down-regulated genes (Figure 5A). GO analysis suggested that these genes were mainly enriched in membrane, fatty acid metabolic process, as well as lipid metabolic process (Figure 5B). KEGG analysis also revealed that fat digestion and absorption was the most significantly enriched pathway (Figure 5C). Using Reactome pathway analysis, most of these genes were found to be associated with lipids and triglyceride metabolism (Figure 5D). ELISA assay further confirmed that DSS treatment significantly increased serum concentrations of triglyceride and low-density lipoprotein while it decreased high-density lipoprotein in WT mice (Figure 5E,G,H). However, no significant changes in these serum lipid parameters were observed in miR-199a-deficient mice.

### 3.6. Validation of Ndrg1 as a Candidate Target of miR-199a-3p

There were seven overlapping genes screened out between 1914 putative target genes of miR-199a-5p and 154 up-regulated genes in miR-199a-deficient mice with colitis, and six overlapping genes between 1531 target genes of miR-199a-3p and the 154 up-regulated genes (Figure 6A). Ndrg1 was considered as a potential candidate for further investigation, and the binding site of miR-199a-3p in the Ndrg1 3′UTR was shown in Figure 6A. After confirming the up-regulation of Ndrg1 in colon tissues (Figure 6B), the ability of miR-199a-3p to regulate Ndrg1 expression by targeting its binding sites was examined using luciferase reporter assay. As demonstrated in Figure 6C, miR-199a-3p mimics dramatically suppressed the luciferase activity of the Ndrg1 3′UTR vector (*p* < 0.01), while mutation of the binding site abolished this repression.

## 4. Discussion

Early weaning stress impairs intestinal epithelial barrier causing immune system dysfunction and growth retardation [25], while profiles and roles of miRNAs in piglets during weaning stress are not yet clear. In this study, the highest levels of miR-199a-5p in jejunal villus cells were observed on day 14 after weaning while the lowest levels in crypt cells were noted on day 7 and 14 after weaning. In contrast, miR-199a-3p was down-regulated in both jejunal villus cells and crypt cells on day 7 after weaning. These findings were consistent with our previous report, which demonstrated that miR-10b, another member of the miRNA family, had a decreased expression after weaning [24]. Moreover, both miR-199a-5p and miR-199a-3p were found to be differently expressed along the villus–crypt axis, which agreed with a previous study demonstrating that miR-199a-5p and -3p were up-regulated in jejunal villus cells as compared with crypt epithelial cells [21]. Given the distinct alterations in miR-199a expression, it may have the potential to be used as a monitoring marker for weaning-associated disorders.

Controlling intestinal inflammation is pivotal in managing gut disorders in piglets after weaning [26]. In the context of inflammation, miRNAs play critical roles in regulating immune processes while ensuring a rapid return to homeostasis [27]. Therefore, a loss-of-function study was conducted to depict the functional roles of miR-199a in the maintenance of immune homeostasis. Here, we showed that mice deficient in miR-199a were dramatically sensitive to the DSS challenge, characterized by decreased survival, dramatic weight change, greater disease activity index, significant colon shortening, as well as more severe tissue damage following DSS challenge. Moreover, the increased morbidity and mortality were correlated with enhanced inflammatory cell infiltration and pro-inflammatory cytokine expression in miR-199a-deficient mice after DSS challenge. These results were in line with those of a previous study, in which miR-199a-3p was shown to regulate the recruitment of immune cells by targeting CD44, thereby ameliorating the severity of DSS-induced colitis [20]. In addition, Peng et al. [28] revealed that miR-199a-3p acted as a negative inflammatory regulator in cervical epithelial cell inflammation via modulating the TLR4/NF-κB signaling pathway. Consistently, Yu et al. [29] also demonstrated a protective effect of miR-199a-5p on neural cells via attenuating inflammation induced by endoplasmic reticulum stress. In this study, the aggravated severity of disease was also correlated with disruption of the intestinal epithelium integrity in miR-199a KO mice, coincident with elevated serum permeability-related parameters. Consistent with our results, lack of miRNA-processing enzyme Dicer1 has also been shown to disrupt intestinal barrier function and aggravate intestinal inflammation with lymphocyte and neutrophil infiltration [30]. Together, these findings demonstrated that miR-199a ablation significantly exacerbated immune response and intestinal barrier dysfunction after DSS treatment, suggesting a protective anti-inflammatory role of miR-199a in the modulation of intestinal inflammation. However, whether restoring the abundance of miR-199a could alleviate DSS-induced colitis needs to be further explored.

Key molecules and pathways implicated in the pathogenesis of colitis in mice lacking miR-199a were further identified. Interestingly, pathway enrichment analysis by GO, KEGG and Reactome revealed that the differentially expressed genes were mainly associated with lipids and triglyceride metabolism. These findings were further confirmed by our ELISA assay showing that DSS treatment increased serum concentrations of triglyceride and low-density lipoprotein while it decreased high-density lipoprotein in WT mice, but not in miR-199a KO mice. In line with our results, a recent study reported that colitis was associated with aberrant hepatic lipid metabolism and adipose tissue dysfunction in mice [31]. In that research, the impaired intestinal permeability and consequent endotoxemia induced by colitis were related to excessive hepatic fat deposition and abnormal lipid profiles. In another study, Li et al. [32] reported that administration of exosomes loaded with miR-199a-5p accelerated liver lipid accumulation accompanied by modulation of hepatic lipogenesis and lipolysis, while delivery of exosomes containing anti-miR-199a-5p ameliorated steatosis in mice fed on high-fat diet. Together with our current work, these findings suggested that miR-199a might be a potential regulator for lipid metabolism during inflammation.

The N-myc down-regulated gene 1 (Ndrg1), a member of the NDRG family, has been implicated in numerous biological processes, including cell proliferation, differentiation, and cellular stress response [33,34]. Evidence has shown that Ndrg1 may play important roles in the progression of inflammatory bowel disease and may be used as a promising therapeutic candidate for the control of intestinal inflammation [35]. In the present study, Ndrg1 was up-regulated in miR-199a-deficient mice, and it was further confirmed as a direct target for miR-199a-3p. Accordingly, the potential connection between miR-199a and Ndrg1 pathway may be related to the development of colitis. Moreover, Ndrg1 has also been recognized as a major regulator of lipid fate in breast cancer cells [36]. Silencing Ndrg1 has been demonstrated to disturb lipid metabolism by enhancing fatty acid incorporation into neutral lipids, while Ndrg1 expression inhibited lipid droplet formation [36]. Given the altered lipid profile observed in mice with colitis, it is worth investigating whether miR-199a/Ndrg1 pathway can modulate lipogenesis and lipolysis during colitis in future studies.

## 5. Conclusions

Taken together, these findings demonstrated an altered expression profile of miR-199a in piglets during the suckling-to-weaning transition and a regulatory role for miR-199a in intestinal inflammation. Enrichment analysis further revealed a potential function of miR-199a associated with lipid metabolism and Ndrg1 was verified as a direct target of miR-199a-3p. Future studies can further examine the anti-inflammatory potential of miR-199a to attenuate weaning-associated intestinal inflammation for livestock.

## Figures and Tables

**Figure 1 animals-13-01979-f001:**
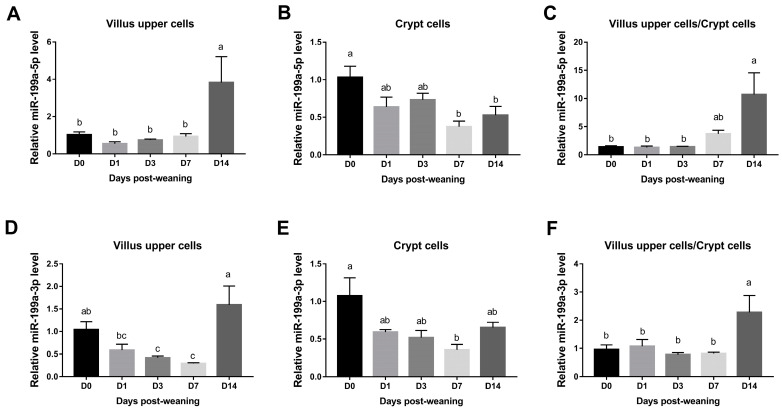
Expression levels of miR-199a in jejunum of piglets during the postweaning period. Relative expression of miR-199a-5p (**A**–**C**) and miR-199a-3p (**D**–**F**) was examined at different time points after weaning. Means without a common superscript differ significantly (*p* < 0.05), *n* = 4.

**Figure 2 animals-13-01979-f002:**
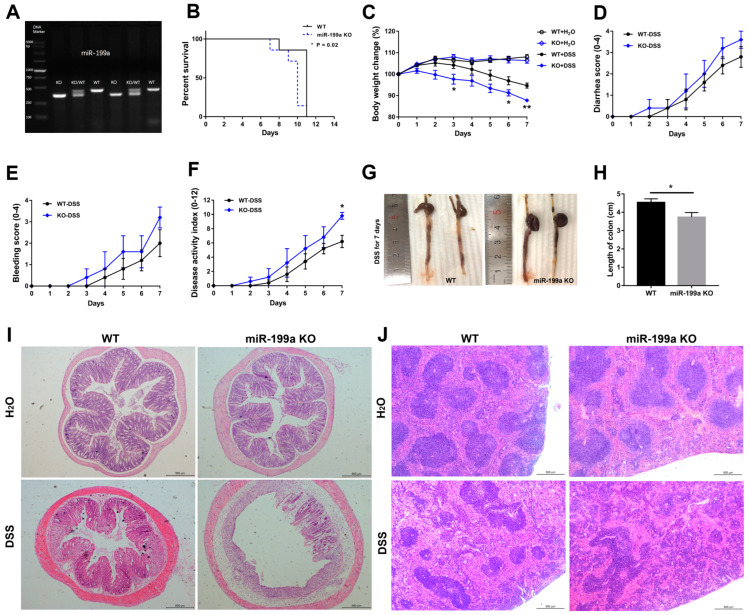
Lack of miR-199a increased susceptibility to DSS-induced colitis. (**A**) Mice genotype was identified by PCR. (**B**) Survival curve (*n* = 7/group; *, *p* < 0.05). (**C**) Body weight loss (*n* = 6/group; *, *p* < 0.05; **, *p* < 0.01). (**D**) Diarrhea score (0–4; *n* = 5/group). (**E**) Bleeding score (0–4; *n* = 5/group). (**F**) Disease activity index (0–12; *n* = 5/group; *, *p* < 0.05). (**G**,**H**) Colon length (*n* = 6/group; *, *p* < 0.05). Typical histological images of (**I**) colon and (**J**) spleen of mice (scale bar, 500 μm).

**Figure 3 animals-13-01979-f003:**
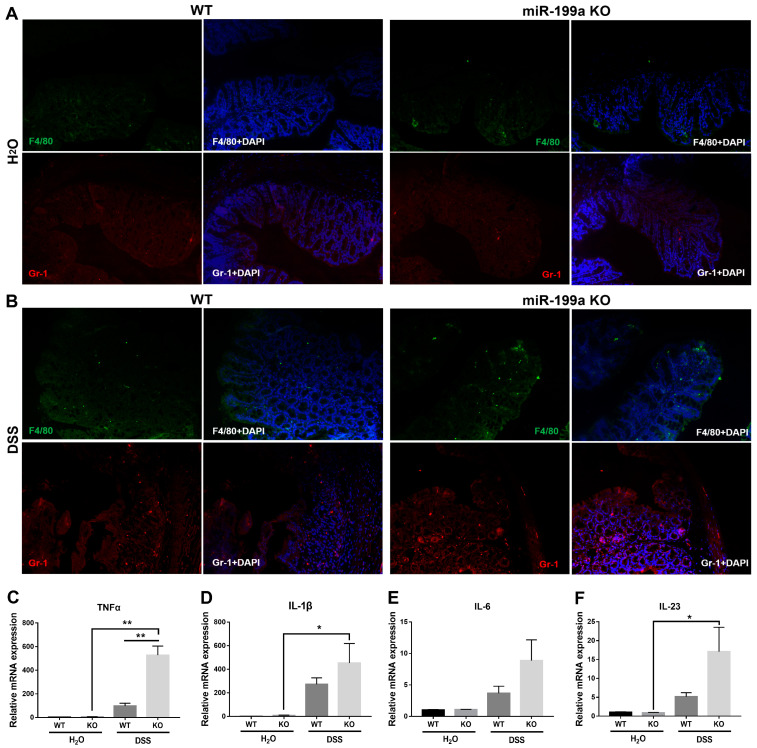
Deficiency of miR-199a exacerbated immune response in mice with DSS challenge. (**A**,**B**) Immunofluorescence for F4/80 (green) and Gr-1 (red) (magnification, ×200; scale bar, 100 μm). Relative expression levels of (**C**) TNF-α, (**D**) IL-1β, (**E**) IL-6, and (**F**) IL-23 in colonic tissues of mice (*n* = 5/group; *, *p* < 0.05; **, *p* < 0.01).

**Figure 4 animals-13-01979-f004:**
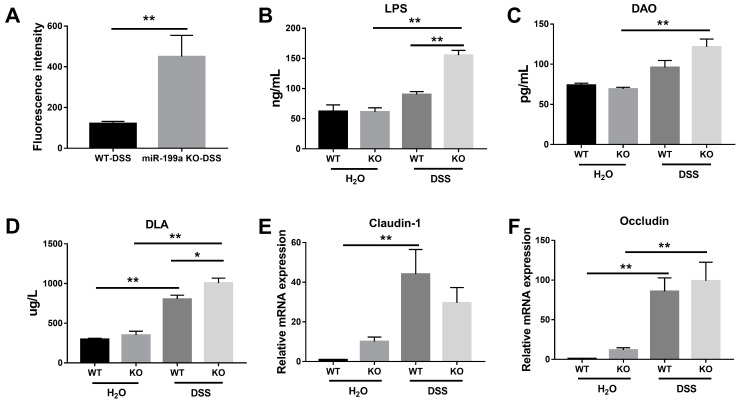
Deletion of miR-199a resulted in increased intestinal permeability in mice with DSS treatment. (**A**) FITC-dextran assay (*n* = 7/group; **, *p* < 0.01). (**B**) LPS, (**C**) DAO, and (**D**) D-LA in serum were measured by ELISA techniques (*n* = 5/group; *, *p* < 0.05; **, *p* < 0.01). Relative expression of (**E**) Claudin-1 and (**F**) Occludin in colonic tissues of mice (*n* = 5/group; **, *p* < 0.01).

**Figure 5 animals-13-01979-f005:**
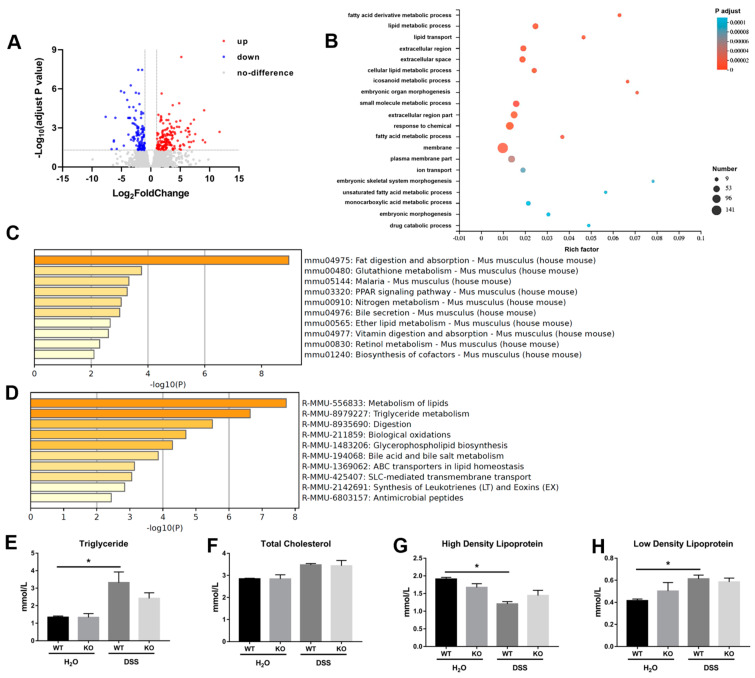
Identification of key pathways in miR-199a-deficient mice with colitis. (**A**) Differentially expressed genes were visualized by volcano plot. (**B**) GO enrichment analysis. (**C**) The top 10 significant enriched KEGG terms. (**D**) The top 10 significant enriched Reactome pathways. Serum concentrations of (**E**) triglyceride, (**F**) total cholesterol, (**G**) high-density lipoprotein, and (**H**) low-density lipoprotein of mice (*n* = 4/group; *, *p* < 0.05).

**Figure 6 animals-13-01979-f006:**
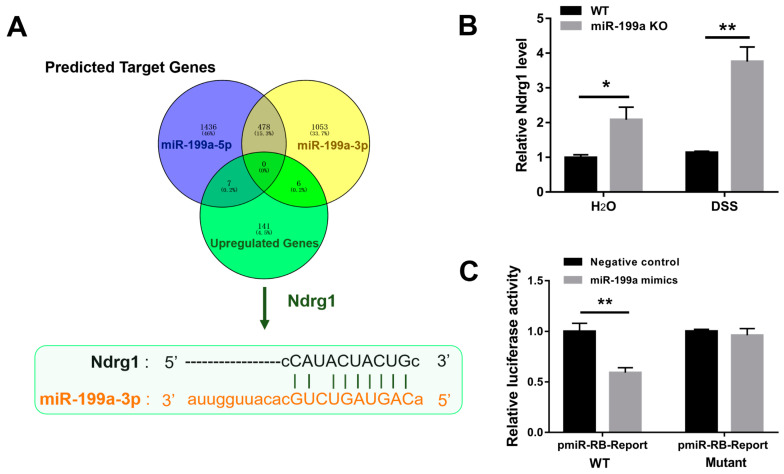
Validation of Ndrg1 as a direct target of miR-199a. (**A**) Ndrg1 was predicted as a possible target gene. Sequence diagram indicating the binding site of miR-199a-3p in the Ndrg1 3′UTR. (**B**) Relative expression of Ndrg1 in colonic tissues (*n* = 5/group; *, *p* < 0.05; **, *p* < 0.01). (**C**) Luciferase activity (*n* = 6/group; **, *p* < 0.01).

**Table 1 animals-13-01979-t001:** Primers used for qRT-PCR analysis.

Genes	Primers	Sequences (5′-3′)	Tm (°C)	GC (%)	Reference
TNF-α	Forward	CCCTCACACTCAGATCATCTTCT	59	48	NM_013693.3
	Reverse	GCTACGACGTGGGCTACAG	60	63	
IL-1β	Forward	ACCTGTCCTGTGTAATGAAAGACG	61	46	NM_008361.4
	Reverse	TGGGTATTGCTTGGGATCCA	59	50	
IL-6	Forward	GCTTAATTACACATGTTCTCTGGGAAA	60	37	NM_031168.2
	Reverse	CAAGTGCATCATCGTTGTTCATAC	59	42	
IL-23	Forward	CACCTCCCTACTAGGACTCAGC	61	59	NM_031252.2
	Reverse	TGGGCATCTGTTGGGTCT	58	56	
Claudin-1	Forward	ACTCCTTGCTGAATCTGAACAGT	60	43	NM_016674.4
	Reverse	GGACACAAAGATTGCGATCAG	58	48	
Occludin	Forward	ACTGGGTCAGGGAATATCCA	57	50	NM_008756.2
	Reverse	TCAGCAGCAGCCATGTACTC	60	55	
Ndrg1	Forward	CATTTTGCTGTCTGCCATG	55	47	NM_008681.2
	Reverse	CCATGCCAATGACACTCTTG	57	50	
β-actin	Forward	GTGCTATGTTGCTCTAGACTTCG	59	48	NM_007393.5
	Reverse	ATGCCACAGGATTCCATACC	57	50	

## Data Availability

All data will be available from the corresponding author upon reasonable request.

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
