# Peer review of "Expression Profile of miR-199a and Its Role in the Regulation of Intestinal Inflammation"

_animals, 2023, doi:10.3390/ani13121979_

Round 1

Reviewer 1 Report

Wu et al investigated the important role of miR-19a in regulation of intestinal inflammation in piglets during weaning period. The finding from this extensive study supports the pivotal role of miR-19a in controlling inflammation by regulating expression of immune genes, gut barrier functions and maintaining intestinal homeostasis. The finding of this study is vital for swine management during the weaning period, which is generally associated with intestinal epithelium injury and stress.

The authors analyzed all the possible parameters related to the intestinal inflammation including growth and physiological parameters, histology, barrier functions, serum biochemical analysis, RNA-seq and qPCR. They presented the results clearly and the manuscript is well written. The conclusion is well supported by the finding of the study.

I have some minor concerns as indicated below.

Methodology section:

qPCR analysis

Table 1. Primer sequences used for qRT-PCR analysis.

Please mention if the primers are designed in the current study or referred from previous studies.

Please include respective references if they are from previous studies.

If they are designed in the current study, please include the efficiency of the primers and reference (NCBI or ensemble or etc..ids) of the respective genes in the table.

Author Response

Response: We appreciate the reviewer’s positive evaluation of our work. The primers used in the present study were designed using the Primer-BLAST tool (https://www.ncbi.nlm.nih.gov). We have revised the Methodology Section and Table 1 according to your valuable advice, on page 3, lines 119-121, and on page 4, lines 124-126.

Reviewer 2 Report

General comments:

This is an interesting manuscript that demonstrates the possible use of micro RNAs as biomarkers for intestinal inflammation, mainly in the post weaning pig. The general concept of the study is clear, but there are some aspects, mainly related to the material and method, that were not very clear. The main point is related to the demonstration that the piglet intestinal cells were in fact derived from the villi or the crypt. I understand that Wang et al (2020) protocol was used for the collection of upper villous and crypt cells, but it would be important to demonstrate that this procedure was effective in its strategy.

Specific comments:

Line 133: The histologic scoring system was based on Wu et al (2023) publication, however, it would be important to state that even briefly for the benefit of the reader.

Lines 145-146: The antibodies used for the immunofluorescent staining are just mentioned in the material and method section, but there is no explanation related to the target of them. It would be interesting to state which cells they are targeting.

Line 205: There is no comment related to the related to the significant high relative level of miR-199a-3p on day 14 in figure 1D.

Line 220: What does worse survival mean? Was it significant different among groups?

Lines 220-221: What does dramatic weight loss mean? Was it significant different among groups?

Figures 2D and 2E are demonstrated in the manuscript but not used or cited in the text.

Lines 225-226: “… with significantly decreased white pulp and increased red pulp…” Was any statistical analysis performed? Why is it significant?

Lines 232-233: Histologic images in Figures 2I deserve a detailed description of the lesions detected in the intestine. It could be done as mentioned for the spleen in the text, or in the legend of the figure.

Author Response

General comments:

This is an interesting manuscript that demonstrates the possible use of micro RNAs as biomarkers for intestinal inflammation, mainly in the post weaning pig. The general concept of the study is clear, but there are some aspects, mainly related to the material and method, that were not very clear. The main point is related to the demonstration that the piglet intestinal cells were in fact derived from the villi or the crypt. I understand that Wang et al (2020) protocol was used for the collection of upper villous and crypt cells, but it would be important to demonstrate that this procedure was effective in its strategy.

Response: We are very grateful for your careful work and thoughtful suggestions. In a previous study reported by our group, Dr. Xiong and Dr. Yang have isolated three cell fractions and validated the fractionation efficiency by alkaline phosphatase (the marker of absorptive cell differentiation). The specific activity of alkaline phosphatase increased 5.6-fold from 1.11 (F3) to 6.20 U/g protein (F1). And we have added this reference to the current study.

Xiong X, Yang H, Tan B, Yang C, Wu M, Liu G, et al. Differential expression of proteins involved in energy production along the crypt-villus axis in early-weaning pig small intestine. Am J Physiol Gastrointest Liver Physiol. 2015;309:229-37.

Specific comments:

Line 133: The histologic scoring system was based on Wu et al (2023) publication, however, it would be important to state that even briefly for the benefit of the reader.

Response: We are grateful for the suggestion and we have described the scoring system briefly as suggested, on page 4, lines 139-142.

Lines 145-146: The antibodies used for the immunofluorescent staining are just mentioned in the material and method section, but there is no explanation related to the target of them. It would be interesting to state which cells they are targeting.

Response: Thank you for your suggestion. We have stated the cell type according to your valuable advice, on page 5, lines 154-156.

Line 205: There is no comment related to the related to the significant high relative level of miR-199a-3p on day 14 in figure 1D.

Response: Thank you for your careful review. We have added comment accordingly, on page 6, lines 215-216.

Line 220: What does worse survival mean? Was it significant different among groups?

Response: Thank you very much for your careful review. We have added P value accordingly, on page 7, line 230.

Lines 220-221: What does dramatic weight loss mean? Was it significant different among groups?

Response: Thank you for your careful review. We have added P value accordingly, on page 7, line 231.

Figures 2D and 2E are demonstrated in the manuscript but not used or cited in the text.

Response: Thank you for your comments. We have cited Figures 2D and 2E on page 7, lines 233-234.

Lines 225-226: “… with significantly decreased white pulp and increased red pulp…” Was any statistical analysis performed? Why is it significant?

Response: Thank you for your careful review. We evaluated the inflammation-associated tissue damage in colon (local inflammatory site) and spleen (systemic inflammatory site) in the present study. The histological scoring system for splenic inflammation was shown in Supplementary Table S1 and the statistical analysis result was demonstrated in Supplementary Figure S3.

Lines 232-233: Histologic images in Figures 2I deserve a detailed description of the lesions detected in the intestine. It could be done as mentioned for the spleen in the text, or in the legend of the figure.

Response: Thank you for your careful review. We have described the lesions detected in the intestine as suggested, on page 7, lines 235-237.

Reviewer 3 Report

Generally, this manuscript is well-prepared and the results provide functional roles of miR-199a in intestinal inflammation, and the findings clearly support the conclusion and extend our knowledge. However, there are still some concerns which need to be addressed:

1.  The quality of most figures needs to be improved.

2. Line 199-200: Please rephrase this sentence.

3. Line 121: Describe the acronym U6 at its first use.

4. Line 138-139: Add the information of manufacturer.

5. Line 142: Add the information of manufacturer.

6. Some words should be written in italic type, such as “P” and “et al.”.

7. Figure 1:The first letter of each graph is capitalized.

8. Conclusions: This part should summarize the conclusions of the manuscript in more refined language.

9. Figure 3 C, D, E, F and Figure 4E, F should be described that  the relative mRNA expression .....

English throughout the manuscript should be improved.

Author Response

Generally, this manuscript is well-prepared and the results provide functional roles of miR-199a in intestinal inflammation, and the findings clearly support the conclusion and extend our knowledge. However, there are still some concerns which need to be addressed:

  1. The quality of most figures needs to be improved.

Response: We are very grateful for your careful work and thoughtful suggestions. We have improved our figures as suggested.

  1. Line 199-200: Please rephrase this sentence.

Response: Thank you for your suggestion. We have rephrased this sentence on page 6, line 210.

  1. Line 121: Describe the acronym U6 at its first use.

Response: We have described the acronym U6 as suggested, on page 4, line 123.

  1. Line 138-139: Add the information of manufacturer.

Response: We have added the information of manufacturer as suggested.

  1. Line 142: Add the information of manufacturer.

Response: We have added the information of manufacturer as suggested.

  1. Some words should be written in italic type, such as “P” and “et al.”.

Response: Thank you for your careful review. We have made correction as suggested.

  1. Figure 1:The first letter of each graph is capitalized.

Response: Thank you for your suggestion. We have revised Figure 1 accordingly.

  1. Conclusions: This part should summarize the conclusions of the manuscript in more refined language.

Response: Thanks for your comments on our paper. We have reworded the Conclusion Section according to your valuable advice, on pages 12-13, lines 390-396.

  1. Figure 3 C, D, E, F and Figure 4E, F should be described that “ the relative mRNA expression .....

Response: Thank you for your careful review. We have revised Figure 3 and 4 as suggested.

Comments on the Quality of English Language

English throughout the manuscript should be improved.

Response: Thank you for your careful review. We have improved the writing of the manuscript accordingly.
